# The Mean of Milk: A Review of Human Milk Oligosaccharide Concentrations throughout Lactation

**DOI:** 10.3390/nu13082737

**Published:** 2021-08-09

**Authors:** Buket Soyyılmaz, Marta Hanna Mikš, Christoph Hermann Röhrig, Martin Matwiejuk, Agnes Meszaros-Matwiejuk, Louise Kristine Vigsnæs

**Affiliations:** 1DSM Nutritional Products Ltd., Kogle Alle 4, 2970 Hørsholm, Denmark; Marta.Miks@dsm.com (M.H.M.); Christoph.Roehrig@dsm.com (C.H.R.); Martin.Matwiejuk@dsm.com (M.M.); Agnes.Matwiejuk@dsm.com (A.M.-M.); louise.vigsnaes@helgum.dk (L.K.V.); 2Faculty of Food Science, Food Biochemistry, University of Warmia and Mazury in Olsztyn, Plac Cieszynski 1, 10-726 Olsztyn, Poland; 3Department of Technology, Faculty of Health, University College Copenhagen, DK-2200 Copenhagen, Denmark

**Keywords:** human milk oligosaccharides, human milk carbohydrates, HMO composition, non-digestible carbohydrates, 2′-fucosyllactose

## Abstract

Human milk oligosaccharides (HMOs) are non-digestible and structurally diverse complex carbohydrates that are highly abundant in human milk. To date, more than 200 different HMO structures have been identified. Their concentrations in human milk vary according to various factors such as lactation period, mother’s genetic secretor status, and length of gestation (term or preterm). The objective of this review is to assess and rank HMO concentrations from healthy mothers throughout lactation at a global level. To this aim, published data from pooled (secretor and non-secretor) human milk samples were used. When samples were reported as secretor or non-secretor, means were converted to a pooled level, using the reported mean of approximately 80/20% secretor/non-secretor frequency in the global population. This approach provides an estimate of HMO concentrations in the milk of an average, healthy mother independent of secretor status. Mean concentrations of HMOs were extracted and categorized by pre-defined lactation periods of colostrum (0–5 days), transitional milk (6–14 days), mature milk (15–90 days), and late milk (>90 days). Further categorizations were made by gestational length at birth, mother’s ethnicity, and analytical methodology. Data were excluded if they were from preterm milk, unknown sample size and mothers with any known disease status. A total of 57 peer-reviewed articles reporting individual HMO concentrations published between 1996 and 2020 were included in the review. Pooled HMO means reported from 31 countries were analyzed. In addition to individual HMO concentrations, 12 articles reporting total HMO concentrations were also analyzed as a basis for relative HMO abundance. Total HMOs were found as 17.7 g/L in colostrum, 13.3 g/L in transitional milk, and 11.3 g/L in mature milk. The results show that HMO concentrations differ largely for each individual HMO and vary with lactation stages. For instance, while 2′-FL significantly decreased from colostrum (3.18 g/L ± 0.9) to late milk (1.64 g/L ± 0.67), 3-FL showed a significant increase from colostrum (0.37 g/L ± 0.1) to late milk (0.92 g/L ± 0.5). Although pooled human milk contains a diverse HMO profile with more than 200 structures identified, the top 10 individual HMOs make up over 70% of total HMO concentration. In mature pooled human milk, the top 15 HMOs in decreasing order of magnitude are 2′-FL, LNDFH-I (DFLNT), LNFP-I, LNFP-II, LNT, 3-FL, 6′-SL, DSLNT, LNnT, DFL (LDFT), FDS-LNH, LNFP-III, 3′-SL, LST c, and TF-LNH.

## 1. Introduction

Breastfeeding has been associated with lower rates of infectious diseases and infantile mortality, reduced risk for obesity, cardiovascular disease, inflammatory bowel disease (IBS), and type II diabetes [1,2,3]. In addition to the milk macronutrients such as lactose, lipids, and proteins, human milk contains complex carbohydrate structures known as human milk oligosaccharides (HMOs), which attract significant attention since they constitute the largest remaining compositional difference between breastmilk and infant formula.

HMOs constitute the third most abundant solid component of human milk, exceeding the amount of protein [4]. Compared to the milk oligosaccharide fraction of human milk, some remarkable differences are observed in the milk oligosaccharide fraction of other mammals: (a) the total amount of milk oligosaccharides are typically significantly lower [5,6], particularly for domestic farmed animals [7]; (b) the structural complexity and bias of individual structures is lower and different [8,9,10]; and (c) different carbohydrate epitopes are detected [11,12]. In addition to their unique abundance in human milk, there is evidence that HMOs are present in the maternal serum during early gestation, in the umbilical cord blood, and in amniotic fluid [13,14,15].

HMOs are largely indigestible by the infant, hence do not function as direct energy resources for the infant. The majority reach the colon where they are utilized by specific infant gut bacteria and approximately 1% is absorbed [16,17]. The HMO profile of human milk appears fundamental for shaping the gut microbiota of the infant by selectively stimulating the growth of specific bacteria, especially bifidobacteria [18]. The bifidogenic effects found in breastfed infants include proliferation of specific bifidobacteria such as *Bifidobacterium infantis*, *B. breve*, and *B. bifidum*, whose genomes encode a large proportion of oligosaccharide processing and transporting genes clustered within conserved loci [19,20,21,22]. The ability of these bifidobacterial species to utilize HMOs implies a co-evolution, where the glycans produced by the host have served as a carbon and energy source for these bifidobacterial species [23]. Bifidobacterial abundance have been linked to host protection from pathogenic bacteria by helping prepare the mucosal immune system, consequently decreasing susceptibility to various diseases later in life [24]. In addition to their bifidogenic activity, HMOs have also shown to be able to directly or indirectly affect mucosal and systemic immunity, help reduce the risk of pathogenic infection and may support brain development in infants [25,26,27,28,29].

### 1.1. Structures and Abbreviations of HMOs

All HMOs derive from lactose (galactosyl-β1-4-glucose) and the HMO fraction of human milk is characterized by extension with four monosaccharides: *N*-acetyl-D-glucosamine (GlcNAc), D-galactose (Gal), sialic acid (Neu5Ac), and/or L-fucose (Fuc). GlcNAc and galactose are added in specific order and linkages to form the neutral-core structures. While Neu5Ac and Fuc can be present on the terminal positions of either lactose or the core structures, forming sialylated and fucosylated groups. The resulting composition of the HMO fraction is complex and diverse with more than 200 different HMO structures detected by sensitive mass spectrometry [30,31,32]. Names, abbreviations, and structures of the most abundant HMOs reported in this review are listed in Table 1. The collective body of analytical data shows that HMOs can be classified into three fundamental structure classes: (1) neutral-core HMOs (containing GlcNAc); (2) neutral fucosylated HMOs (containing fucose); and (3) acidic HMOs (acidic fucosylated and acidic nonfucosylated) (containing sialic acid). Table 2 lists the 15 most abundant structures reported in this review. For further information on the chemical structures of individual HMOs, it is recommended to refer to Chen et al. (2015), who provide an elaborate resource to discover the structures of HMOs [32].

### 1.2. Factors Influencing HMO Variability

The composition and concentration of individual HMO structures in human milk vary according to genetic and non-genetic factors. Even though neither carbohydrate nor oligosaccharide synthesis is directly genetically encoded (unlike DNA, RNA, and proteins), HMO variability is strongly dependent on the activity of two specific enzymes that are encoded by the Secretor (*Se*) and Lewis (*Le*) genes in the mother [27,29]. *Se* and *Le* genes encode the enzymes α1-2-fucosyltransferase (FUT2) and α1-3/4-fucosyltransferase (FUT3) respectively, both affecting the biosynthesis of fucosylated HMOs [27]. Milk from women with inactivated FUT2—due to a diversity of different mutations in both alleles of the *Se* gene—contain zero or only traces of the α1-2-fucosylated HMOs. Similarly, FUT3 activity can be inactivated (or severely reduced) due to a diversity of different mutations in both alleles of the *Le* gene, thus Lewis-negative women’s milk contain zero or only traces of the α1-4-fucosylated HMOs (α1-3-fucosylation is encoded by several enzymes and is therefore not eliminated by FUT3 deactivation). Therefore, mothers who carry the *Se* gene express FUT2 enzyme while mothers who do not carry *Se* do not express the FUT2 enzyme. The interplay of the FUT2 and FUT3 enzymes leads to two main types of Lewis antigens (Le^a^ and Le^b^) and four common milk phenotypes are observed: Se+/Le^(a−b+)^, Se-/Le^(a+b−)^, Se+/Le^(a−b−)^, Se−/Le^(a−b−)^. Reflecting these phenotypic differences, HMO profiles in lactating mothers are classified into four milk phenotypes, or four different ‘milk groups’ resulting in distinct structural features in their oligosaccharide fraction (Table 1) [33,34]. There are also different consequences of each milk group for the infant postulated. For instance, moderate-to-severe and calicivirus-associated diarrhea occurred less often in infants whose milk contained high levels of 2-linked fucosylated HMOs, suggesting that HMO profile is clinically relevant for incidence of diarrhea [35].

The highest HMO concentrations are generally found in colostrum (first milk). Other than lactation period, the non-genetic factors contributing to the quantitative and structural variability of HMOs among mothers are still to be unraveled. There is evidence demonstrating HMO variability across different populations [36,37] which is possibly explained in large part by irregular distribution of milk groups geographically [38]. Non-genetic factors that might contribute to variation in HMO contents are mother’s age, nutritional status, weight, body-mass index, and significant health issues of the mother [39].

In addition to natural causes described above (particularly genetic secretor status and lactation period) differences in reported HMO concentration are certainly also linked to varying inter-laboratory methodologies for glycan analysis [40]. Nevertheless, existing literature of HMO quantification represents a useful resource to determine the mean concentrations of individual HMO structures, and investigate their variation over the course of lactation. A previous systematic review by Thurl et al., provides knowledge on HMO amounts in human milk from mothers of known secretor status [41]. To date, no review has been performed to determine more robustly the most abundant HMOs observed globally in term human milk throughout lactation and their representative mean concentrations. In the present review, each included publication was regarded as an observational point with the central aim to make an estimation of the global HMO profile for pooled samples regardless of the individual variations caused by genetic and non-genetic factors. The authors decided on determining an overview on pooled milk samples instead of secretor and non-secretor samples given that the chosen objective was to determine a ranking of individual HMOs by global means across different milk phenotypes, regardless of individual variations and has included all relevant and reliable HMO concentration data identified.

## 2. Materials and Methods

### 2.1. Literature Search and Selection

Literature search was conducted on seven databases (Biosis, Embase, Medline, PubMed, Web of Science Core Collection, Google Scholar, and Scopus) in the period of March 2019 to December 2020. The search terms included phrases such as “human milk oligosaccharide quantification or concentration”, “breast milk oligosaccharide quantification or concentration”, “human milk oligosaccharide content”, and “human milk carbohydrate concentration” as well as searching the names of individual oligosaccharides (2′-FL, 3-FL, LNT, etc.). The search was also limited to full-text articles in the English language with publication dates later than 1960. Quantitative data availability was the first inclusion criterion. For initial database construction, a total of 89 publications published between 1960 and 2020 were eligible. From these publications, quantification data for 44 HMOs were registered to the database in the form of mean concentration in g/L. The concentration data were assigned the reported sample size, lactation period, region, country, milk type (secretor/non-secretor/pooled), length of gestation (term/preterm), other discriminator (e.g., health status of the mother), and analytical method used for quantification. Pooled milk type indicated that the sample was not categorized according to the mother’s genetic secretor status. For the objective of this review, several assumptions, and approaches were put forward:Milk group: When no information on milk group or milk type were provided (as was the case for four publications), the samples were assumed to be pooled.Secretor/non-secretor data: Milk group-specific means were recalculated according to the typically observed distribution of secretor/non-secretor phenotypes in the population (approx. 80/20%)When HMO amounts were reported with the four specified milk group status, calculations on assumption of a 70:20:9:1 ratio between the four milk groups (see also Table 2 [33]) were applied to mimic a pooled milk sample (calculations explained in Appendix A). If the FUT2 dependent 2′-fucosyllactose (2′-FL), difucosyllactose (DFL), and lacto-*N*-fucopentaose I (LNFP-I) means were collectively <0.5 g/L then the assumption of natural phenotype distribution is almost certainly wrong, and an excess of non-secretors was present in the pool. If such levels were identified, they were excluded.Birth type: When information on mode or type of delivery was not available, it was assumed that birth was term.Lactation period: Since each publication reports differing lactation periods, registered quantification data were assigned to one of the pre-defined lactation periods: 0–5 days (colostrum); 6–14 days (transitional milk); 15–90 days (mature milk); >90 days (late milk). If information on lactation period was not available, data were excluded. In certain cases, data could fit to more than one of the pre-defined lactation periods. For instance, a concentration data for ‘3–90 days’ could fit to several of the above-mentioned lactation periods. In these cases, data were assigned to the lactation period with most days fitting the pre-defined period; in this case 3–90 days was assigned to the 15–90 days period.Other discriminator: Data from mothers with any specified disease conditions were excluded.Data units: HMO means reported in literature were transferred as mean concentration in g/L and sample size. When sample size information was not available, the articles were excluded. If other units than g/L were reported, such as mmol/L the mean concentrations were converted to g/L for consistency. Studies reporting median levels were estimated to mean with standard deviation using a mean variance estimation model [42]. Multiple statistical models are publicly available and in the present review, the model proposed by Wan et al. (2014) was applied.Combined concentrations: Some studies reported combined means of HMOs such as LSTa/b, LNT/LNnT, 2′-FL/3-FL, LNFP-I/II/III, etc. Combined concentrations were transferred to a separate sheet and excluded in the final analysis.

The objective of the review was to reach a global indication of the means of individual HMO structures in healthy term milk. In addition to the first inclusion criterion of HMO quantification, further inclusion/exclusion criteria were defined according to this objective. Data were categorized by the following project-specific inclusion criteria:Pooled samplesTerm milkAll nationalitiesMother without reported disease status.


Accordingly, to the above-listed inclusion criteria, the following data were excluded from the data analysis:
Preterm milk data from eight articles [43,44,45,46,47,48,49,50].Data from one article reporting concentration from Celiac positive mothers [51].Data from one article expressing concentration in relative abundance, which could not be quantitatively assessed [31].Data from three articles due to imprecise nomenclature not allowing a structure assignment [52,53,54,55].Due to unreported sample size, four articles were excluded [54,56,57].Due to unknown lactation period, one article was excluded [58].

In the exclusion process, data from some articles were excluded entirely while for some other articles data fitting the project-specific inclusion criteria were kept. For instance, in some cases data were reported for both term and preterm samples. Therefore, in these cases, the data exclusion did not lower the total article number in the review. The data exclusion process resulted in a total of 57 articles included in the review (Figure 1). After the above-mentioned data exclusion step, the remaining articles were re-assessed. Since ‘pooled samples’ was an inclusion criterion, HMO data for secretor and non-secretor specific milk types were spared for calculation and converted to a pooled sample scenario as described earlier. Data were then re-introduced. Overall, the breast milk samples in the included articles originated from 31 countries (Table 3). Heterogeneity among the included studies manifested as variability among the sample sizes, day of lactation in which the breast milk sample was taken, and the type of analytical method applied for quantifying HMOs.

### 2.2. Statistical Analysis

Data from the constructed database were mean (g/L) and pooled levels of quantified individual HMOs from 57 publications, and total HMOs from 12 publications. The mean concentration data for individual HMOs and total HMOs (already categorized by pre-defined lactation periods) were transferred into GraphPad Prism (version 9.0.1 for Windows, GraphPad Software, San Diego, CA, USA, www.graphpad.com, accesed on 18 August 2020) for statistical analysis. Mean levels which were categorized by lactation period were applied a frequency distribution analysis on GraphPad. A weighted average approach was taken considering the sample size of each article differed (sample sizes were inserted in the N column on GraphPad data sheets). Resulting descriptive statistics were assessed and histograms were built. The results were evaluated in two categories: (i) individual HMOs throughout lactation (analysis of 57 articles), and (ii) total HMOs throughout lactation (analysis of 12 articles).

## 3. Results

### 3.1. Overview of the Review

The review included HMO means from 57 publications with varying characteristics of sample size, lactation period, analytical method and geographical region. All the included articles are cited in Table 3. In the included articles, the largest sample size (number of milk donors) was 2434 while lowest was 1 [67,87,91,97]. The structures and abbreviations of the 15 most abundant HMOs resulting from the present study are shown in Table 1. 

The most common analytical methods used to quantify HMOs were high performance liquid chromatography (HPLC) with fluorescence detection, high-performance anion-exchange chromatography (HPAEC) with pulsed amperometry detection, LC-MS (liquid chromatography with mass spectrometry)/MRM (multiple reaction monitoring), CE (capillary electrophoresis), paper chromatography, NMR (nuclear magnetic resonance), or nano-LC-chip-TOF (time of flight).

The amount of data reported depended on the individual HMO and the lactation period. Table 4 and Table 5 show the number of articles included per lactation period as well as the total sample sizes from the articles. Less data was available for total HMO concentrations while the majority of the included articles reported data for colostrum and mature milk. Even though there were fewer articles reporting the individual HMOs in late milk, total sample sizes were higher compared to colostrum milk which had the greatest number of articles (Table 4).

### 3.2. Concentrations of Individual HMOs in Colostrum (0–5 Days), Transitional (6–14 Days), Mature (15–90 Days), and Late (>90 Days) Milk

Overall, the concentrations of HMOs throughout lactation appear dynamic and structure-dependent. Figure 2 shows the overall fluctuations of the 15 most abundant HMOs resulting from the review. The most abundant individual HMOs resulting from the analysis belong to the fucosylated group and appear in mature human milk as 2′-FL, LNDFH-I (DF-LNT), LNFP-I, LNFP-II, 3-FL, DFL (LDFT), LNFP-III, and TF-LNH. 6′-SL, DSLNT, and 3′-SL appear as the most abundant sialylated HMOs in mature milk and LNT and LNnT as the most abundant neutral core HMOs. 2′-FL is overall the most abundant HMO at all lactation periods. LNT appeared as the most abundant neutral core HMO and 6′-SL appeared as the most abundant sialylated HMO at all lactation stages.

Throughout lactation, 2′-FL significantly decreases from colostrum to transitional, and from mature to late (two-way ANOVA, *p* < 0.0001). LNFP-I and DFL also decreases steadily throughout the defined lactation periods (*p* < 0.0001). LNDFH-I exhibited significant fluctuations with increases and decreases from transitional to late (*p* < 0.0001). On the other hand, 3-FL showed a contrary pattern with a steady increase from colostrum through late milk (*p* < 0.0001).

The resulting descriptive statistics of the analyzed structures in human colostrum showed that 2′-FL is the most abundant HMO at all lactation periods. For colostrum, 2′-FL (mean 3.18 g/L) is followed, in decreasing order, by LNFP-I, LNDFH-I, LNT, 6′-SL, LST c, LDFT (DFL), LNFP-III, LNnT, F-LNH-I, F-LNH-II, DSLNT, 3-FL, LNFP-II, and TF-LNH (Table 6). This list was followed by DS-LNH, F-LNH-III, 3′-SL, DF-LNH a, and others (data shown in Appendix A). In transitional milk, 2′-FL (mean 2.07 g/L) is followed by LNFP-I, LNT, 6′-SL, DSLNT, DF-LNH a, 3-FL, LDFT, LST c, LNnT, F-LNH-II, F-LNH-III, LNFP-III, LNDFH-I, and LNFP-II. For mature milk, 2′-FL (mean 2.28 g/L) is followed by LNDFH-I (DFLNT), LNFP-I, LNFP-II, LNT, 3-FL, 6′-SL, DSLNT, LDFT (DFL), FDS-LNH, LNFP-III, 3′-SL, LST c, and TF-LNH shown in Table 6. In late milk, 2′-FL (mean 1.65 g/L) is followed by 3-FL, LNDFH-I, LNT, LNFP-I, 6′-SL, LDFT (DFL), LNFP-II, LNFP-III, DSLNT, TF-LNH, LNnT, LNDFH-II, 3′-SL, and FDS-LNH. All structures quantified in the four lactation periods are shown in Appendix A.

The individual HMOs resulting from the review were categorized and summed according to their HMO group (neutral core, fucosylated, or sialylated). Figure 3 shows that the fucosylated sum remains the most abundant HMO portion compared to sialylated and neutral core group.

### 3.3. Total HMOs throughout Lactation and Relative Abundance of Individual HMOs

In addition to quantification of individual HMOs, 12 articles published between 1960 and 2020 reporting the amount of the total HMO fraction (not to be mistaken for the sum of the individually analyzed single HMOs) were analyzed (Table 7). Total HMO mean was 17.7 ± 3.3 g/L at colostrum, 13.3 ± 6.5 g/L at transitional, 11.3 ± 2.62 g/L at mature (Table 8). Total HMOs in late milk is not shown due to low number of data points.

With the aim to have an overview of the relative proportions of individual HMOs within the total amount of HMO, two pie chart illustrations were constructed using the data from the mature milk (15–90 days) period. Mature milk was chosen given that this was the period with the most data points both for individual HMOs and for total HMO data. The pie chart was constructed by summing the individual HMOs resulting from the review and taking the total HMO mean for mature milk in Table 8. The ‘all other’ slice was calculated by subtracting the sum of the HMOs from the total HMO mean for mature milk (11.3 g/L). The pie chart illustration showed that, although a large number of individual HMOs exist (≈200 structures), the majority of the whole appears to be made up of a relatively lower number of individual HMOs, indicating that a high number of HMOs are found in relatively low amounts. It appears that the top six make up over 50%, and the top 15, over 75% (Figure 4).

Since biological function is primarily driven by molar concentration a pie-chart was also prepared that looks into the relative molar ratios of HMO abundance (Figure 4). In this representation of the dataset the 10 most abundant HMOs in order of relative concentration include 2′-FL (32%), 3-FL (10%), LNDFH-I (8%), LNT (7%), LNFP-I (7%), LNFP-II (6%), 6′-SL (4%), LNnT (4%), and DFL (3%). It becomes apparent that the metabolic flow-equilibria of the diverse glycosyltransferases of the mammary gland that biosynthesize HMOs from lactose, and that can lead to substantial structural complexity, are rather balanced on the lower molecular weight end of the HMO spectrum (4 out of the 10 most abundant HMOs are smaller trisaccharides and another 3 are tetrasaccharides).

## 4. Discussion

Human milk contains hundreds of bioactive components that aid in the protection of the infant from infection and inflammation as well as contributing to the infant’s immune system, healthy organ development, and microbiota formation [112]. There is considerable difference between the composition of human milk and infant formula [113]. The specific health benefits of human milk as opposed to formula feeding have been recognized for a long time. Particularly, the incidence of infections, diarrhea, allergy, later in life obesity, IBS, and type II diabetes seem to be lower in breastfed infants than in formula-fed infants [3]. Advances in infant formulations aim for a product that is increasingly similar to breast milk in both composition and in outcomes. A more precise understanding of the bioactive components of human milk could thereby lead to improved health of infants and potentially adults. HMOs are largely non-digestible molecules with an abundance and diversity unique to human milk. Individual HMOs have been linked to microbiota modulation, immunomodulation, anti-inflammatory effects, supporting of healthy gut functions, and potential impacts on brain health through the gut–brain axis. 

The aim of the review was to arrive at a better understanding of a representative global HMO composition in term milk from healthy mothers based on mean levels in pooled milk samples, which had not been done before. Published between 1996 and 2020, 57 studies with HMO quantification data have been included. Overall, the results show that HMO concentration is dynamic throughout lactation, with about 15 individual HMO structures constituting the majority (>75%) of the total HMO fraction. This result is in agreement with previously published analytical data showing that a subset of merely 10–15 of HMOs constitute the majority of the total oligosaccharide fraction of human milk [114]. Specifically, 2′-FL, LNDFH-I, LNFP-I, LNFP-II, LNT, 3-FL, 6′-SL, DSLNT, LNnT, DFL, FD-LNH, LNFP-III, 3′-SL, LST c, and TF-LNH appear to constitute the bulk of the total HMO composition of mature milk and the smaller HMOs (≤hexasaccharide) dominate on a relative molar abundance ratio. Given that human milk has been reported to contain approximately 200 different HMO structures, it is relevant to note that a relatively high number of HMO structures are found in rather low amounts compared to the more abundant structures. The results also show that individual HMOs are variable and dynamic throughout lactation.

It is well-established that total HMO and individual structures vary to a great extent in individuals. As mentioned by Thurl et al. (2017), the variability in HMO concentrations is a combination of natural variability and the different quantification methods; however, the effect of these various quantification methods is difficult to test [41]. The results from each of the four defined lactation periods of the study indicate that total HMOs are found in higher amounts during the earliest lactation periods while they decrease after the colostrum period. Most of the individual structures appeared to decrease throughout lactation, although some exceptions exist where certain structures fluctuate, remain stable, or increase at the post-colostrum period. 3-FL significantly increases after colostrum, which was also reported by Borewicz et al. (2020),Wang et al. (2020), and more recently by Plows et al. (2021) and Gu et al. (2021) [69,115,116,117]. Plows et al. (2021), reported that 3-FL increased ten-fold from one month to 24 months [116]. The results of the present review showed that 3-FL increased two-fold from colostrum to three months. 

Limitations of the present study result from the overall data heterogeneity and significant treatment to the raw data according to the set principle to obtain a pooled overview. Data heterogeneity among the included articles manifested as differences in geography, sample size (number of donors), quantification methodology (varying sample treatment and analytic tools used), differing reported data units which required conversion to g/L (nmol/L, mg/L), differing statistical units which required conversion to mean (i.e., median). The mentioned differing characteristics between the included articles may represent a limitation as a substantial amount of data re-handling was performed. 

The present study reported the most abundant HMO structures. The fact that they are most abundant suggests an important biological function, although there should be no reverse implication that the less abundant HMOs are functionally unimportant. Structures in lower abundance may carry important physiological roles for the infant. However, there is lack of functional evidence for the least abundant structures reported in this review. Hence, it is reasonable to assume that biological effect might to some degree rather be a function of structural elements presented than of precise individual HMO structure and a justified approach to aim at reflecting biological effects of the HMO fraction is by covering the structural characteristic elements.

The highest total HMO concentration was found in colostrum (0–5 days), amounting to 17.6 g/L, and decreasing steadily. However, the amount of breast milk consumed by the infant increases after colostrum, through transitional and mature stages. The increased intake of milk and decreased HMO concentrations is expected to have a balancing effect to the HMO amount consumed by the infant. 

Although there is apparent variability in individual structures’ concentrations in human milk, the specific relationship between HMO variability and infant health has been limited to association studies mostly. For instance, HMO content in the milk of Malawian mothers with 6 months old infants suggest that HMOs are significantly lower in the milk of mothers with severely stunted infants [118]. When young germ-free mice colonized with the microbiota of severely stunted infants were fed sialylated milk oligosaccharides, there was a microbiota-dependent increase in lean body mass gain as well as other metabolic improvements [118]. 

It is well-established that the genetic secretor status and various non-genetic factors influence the composition and concentrations of certain HMOs. It should be noted that the resulting mean values in the present study do not represent the bimodal nature of HMOs’ quantitative distribution in nature. Numerous studies have assessed the quantitative variation of HMOs among secretor and non-secretor individuals with a roughly 80/20% rule of secretor and non-secretor distribution. With that in mind, it might appear counter-intuitive to determine a pooled mean instead of observing the bimodal distribution and therefore generating two representative means per HMO (for secretor and non-secretor). With the consideration that no review has been done to investigate the possibility of having a robust overview of the global composite HMO profile of the healthy mother, the authors deliberated a pooled overview.

As mentioned earlier, advances in infant formulation aim for a product that is increasingly like breast milk. It may be unlikely that infant formulations will account for the secretor and non-secretor differences in determining HMO levels in products. Two HMOs, 2′-fucosyllactose (2′-FL) and lacto-N-neotetraose (LNnT) have been most commonly added to infant formula. Healthy term infants fed a 2′-FL and LNnT supplemented formula shifted the microbiota of the test infants closer to the reference breastfed infants with lower parent-reported medication use, respiratory infections and bronchitis [119]. More recently, an infant formula with five HMOs (2′-FL, LNnT, LNT, DFL, 3′-SL, 6′-SL) has become available. Considering that infant formula innovation continuously aims for a product that is compositionally and functionally closer to human milk, understanding the most representative HMO composition of human milk on basis of healthy mothers from all over the globe may facilitate improved formulations. Given that HMOs’ addition in infant formulas and other nutritional compositions will require the approximate values fitting the overall population, the authors decided on a pooled overview. Therefore, regardless of the genetic and non-genetic differences which influence the concentrations of certain HMOs, the present study aimed to illustrate a global picture of the HMO profile in lactating mothers with term born infants. 

## 5. Conclusions

The present study analyzed individual HMO concentrations from 57 articles published between 1996 and 2020 and total HMO concentrations published between 1960 and 2020. Results showed that the concentrations of individual HMOs are variable depending on the lactation period and dynamic throughout lactation. Most individual HMOs exhibited a tendency to decrease or remain stable after colostrum while 3-FL exhibited increase after colostrum. About 15 HMOs constitute more than 75% of the total HMO fraction in mature milk. Overall, 2′-FL, LNDFH-I, LNFP-I, LNFP-II, LNT, 3-FL, 6′-SL, DSLNT, LNnT, DFL, FD-LNH, LNFP-III, 3′-SL, LST c, and TF-LNH appear as the most abundant structures. 

## Figures and Tables

**Figure 1 nutrients-13-02737-f001:**
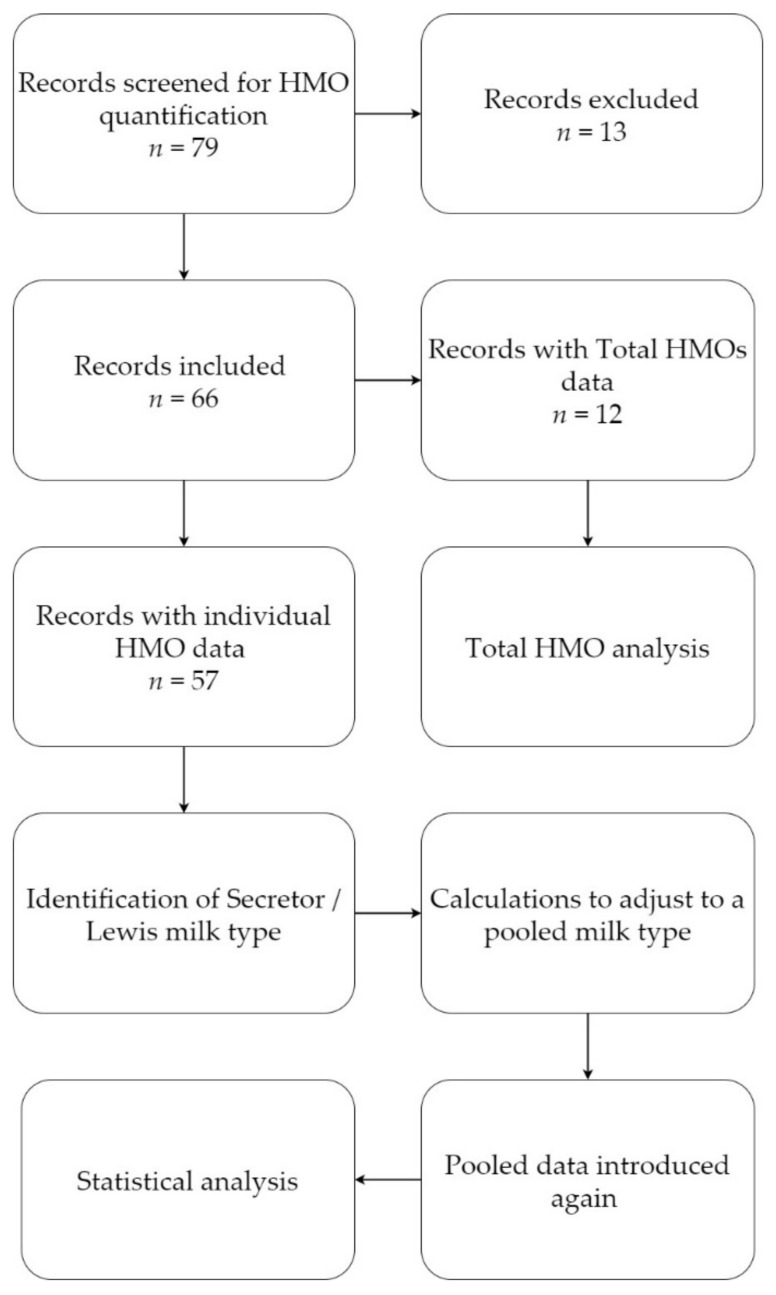
Flow chart of the review.

**Figure 2 nutrients-13-02737-f002:**
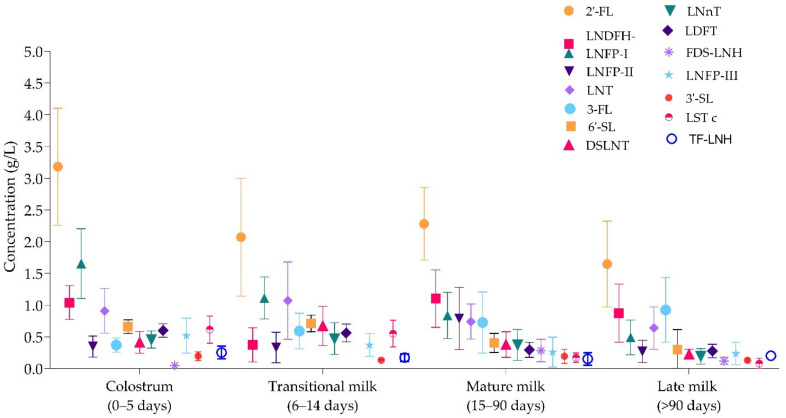
Overview of the 15 most abundant HMOs throughout the pre-defined lactation periods.

**Figure 3 nutrients-13-02737-f003:**
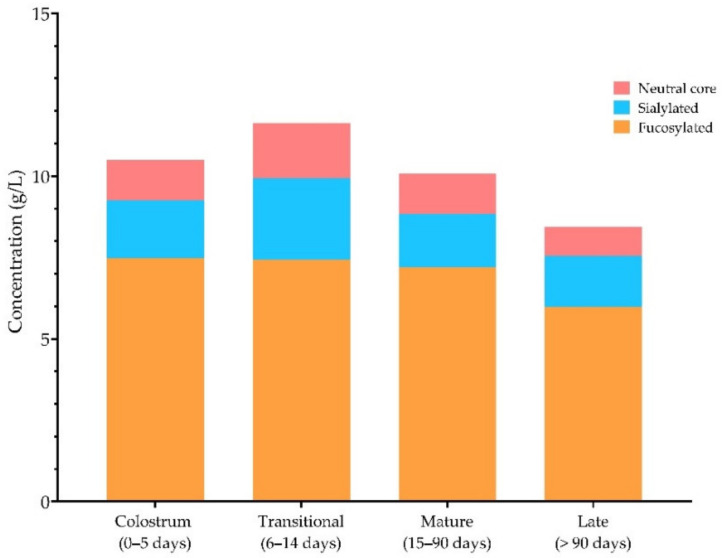
Portions of the neutral core, fucosylated and sialylated HMO groups throughout the pre-defined lactation periods. The Figure was obtained by summing the individual structures in the categories of fucosylated, sialylated, and neutral-core. All the structures are provided in Appendix A.

**Figure 4 nutrients-13-02737-f004:**
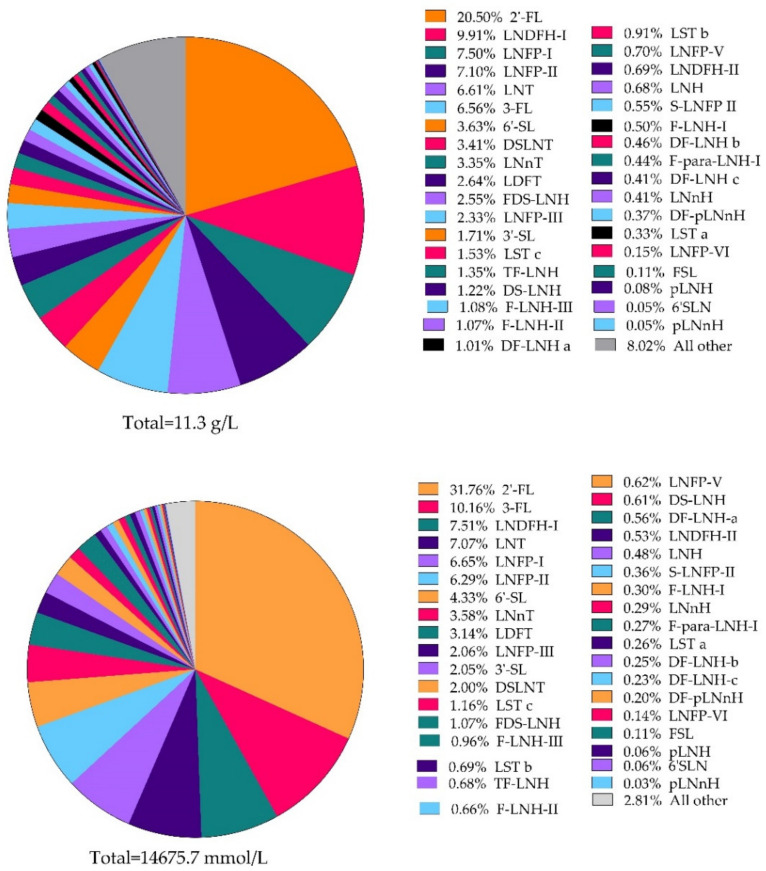
Putative composition of the HMO fraction based on g/L and mmol/L.

**Table 1 nutrients-13-02737-t001:** Structures of the 15 most abundant HMOs in mature human milk reported in this review.

Abbreviation	Name	Structure	Abbreviation	Name	Structure
Neutral HMOs (neutral core and neutral fucosylated)			Acidic non-fucosylated HMOs		
LNT	Lacto-*N*-tetraose	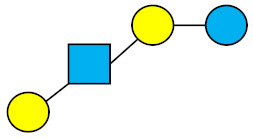	3′-SL	3′-Sialyllactose	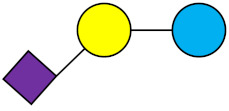
LNnT	Lacto-*N*-neotetraose	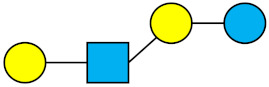	6′-SL	6′-Sialyllactose	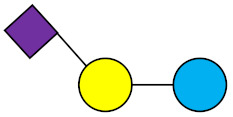
2′-FL	2′-Fucosyllactose	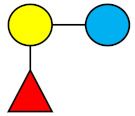	DSLNT	Disialyllacto-*N*-tetraose	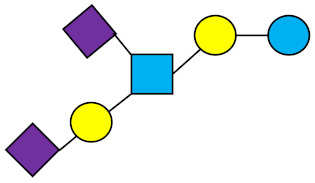
3-FL	3-Fucosyllactose	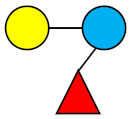	LST c	Sialyllacto-*N*-neotetraose c	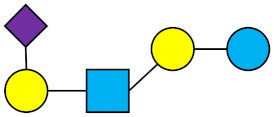
DFL (LDFT)	Difucosyllactose	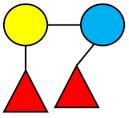		Acidic fucosylated HMOs	
LNDFH-I (DFLNT)	Lacto-*N*-difucohexaose I	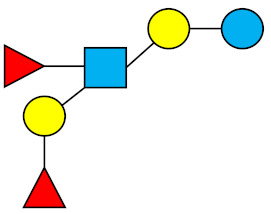	FDS-LNH-I	Fucosyldisialyllacto-*N*-hexaose I	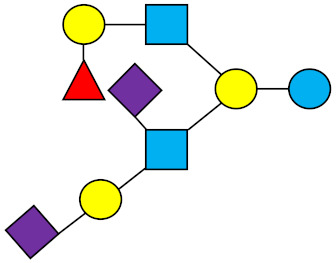
LNFP-I	Lacto-*N*-fucopentaose I	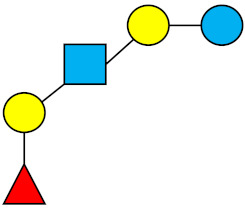	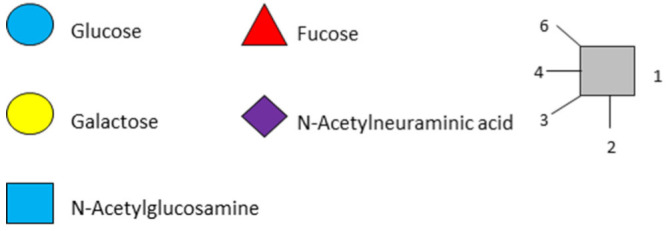		
LNFP-II	Lacto-*N*-fucopentaose II	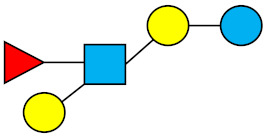			
LNFP-III	Lacto-*N*-fucopentaose III	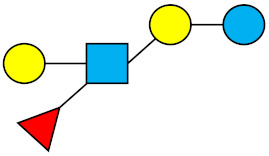			
TF-LNH	Trifucosyllacto-*N*-hexaose	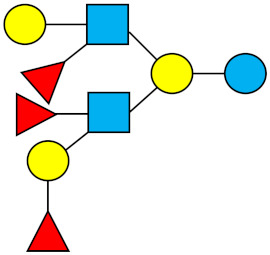			

**Table 2 nutrients-13-02737-t002:** Distribution of mothers’ phenotypes and corresponding milk groups.

Secretor Status	Secretor	Non-Secretor	Secretor	Non-Secretor
Milk group	1	2	3	4
Milk Phenotype	Se+/Le^(a^^−b+)^	Se−/Le^(a+b^^−)^	Se+/Le^(a^^−b−)^	Se−/Le^(a^^−b−)^
α1,2-fucosylated HMOs (FUT2 enzyme ^1^)	+	−	+	−
α1,3-fucosylated HMOs (FUT3, FUT5, FUT6 enzymes)	+	+	+	+
α1,4-fucosylated HMOs (FUT3 enzyme)	+	+	−	−
Typical frequency in global population	70%	20%	9%	1%

^1^ Group 1 (secretor) mothers express both FUT2 and FUT3. Group 2 (non-secretor) mothers express FUT3 but not FUT2. Group 3 (secretor) mothers express FUT2 but not FUT3. Group 4 (non-secretor) mothers express neither FUT2 nor FUT3 [33]. FUT = fucosyltransferase.

**Table 3 nutrients-13-02737-t003:** Characteristics of the included articles reporting individual HMO concentrations in alphabetical order.

Reference	Sample (Number of Donors)	Region	Population	Milk Type	Analytic Method
Aakko et al., 2017 [59]	11	Europe	Finland	Secretor	HPLC
Alderete et al., 2015 [60]	25	North America	USA	Pool	HPLC
Asakuma et al., 2007 [61]	20	Asia	Japan	Pool	HPLC
Asakuma et al., 2008 [62]	12	Europe	Italy	Pool	HPAEC
Asakuma et al., 2011 [63]	57	Asia	Japan	Pool	HPLC
Austin et al., 2016 [64]	446	Asia	China	Pool	HPLC
Austin et al., 2019 [48]	25	Europe	Switzerland	Secretor/Non-secretor	HPLC
Azad et al., 2018 [65]	427	North America	Canada	Secretor/Non-secretor	HPLC
Bao et al., 2007 [66]	10	North America	USA	Pool	CE
Bao et al., 2013 [67]	1	North America	USA	Secretor	HPLC-MS
Borewicz et al., 2019 [68]	121	Europe	Netherlands	Pool	LC-MS
Borewicz et al., 2020 [69]	24	Europe	Netherlands	Pool	UHPLC-MS/HPAEC-PAD
Csernak et al., 2020 [70]	1	Europe	Hungary	Pool	LC-MS
Chaturvedi et al., 1997 [71]	50	Latin America	Mexico	Pool	HPAEC-PAD
Chaturvedi et al., 2001 [72]	12	North America	USA	Secretor/Non-secretor	HPLC
Coppa et al., 1999 [52]	18	Europe	Italy	Secretor	HPAEC-PAD
Coppa et al., 2011 [53]	39	Europe	Italy	Secretor/Non-secretor	HPAEC
Erney et al., 2000 [36]	4–129	Asia, Europe, Latin America, North America	n.a.	Pool	HPAEC
Erney et al., 2001 [73]	368	North America, Europe	n.a.	Pool	HPAEC
Ferreira et al., 2020 [74]	75	Latin America	Brazil	Pool	HPLC
Galeotti et al., 2014 [55]	3	Europe	n.a.	Secretor/Non-secretor	CE-UV
Hong et al., 2014 [75]	20	North America	USA	Secretor/Non-secretor	LC-MS/MS-MRM
Huang et al., 2019 [76]	33	Asia	China	Pool	UHPLC
Kunz et al., 2000 [77]	4	Europe	n.a.	Pool	HPAEC-PAD
Kunz et al., 2017 [78]	21	Europe	Spain	Secretor/Non-secretor	HPAEC-PAD
Lagström et al., 2020 [79]	802	Europe	Finland	Pool	HPLC
Lefebvre et al., 2020 [80]	28-156	Europe	Germany	Pool	LC
Leo et al., 2009 [81]	8	Asia Pacific	Samoa	Pool	HPLC
Leo et al., 2010 [82]	16	Asia Pacific	Samoa	Pool	HPLC
Ma et al., 2018 [83]	20	Asia	China, Malaysia	Pool	HPLC-MS
Martin-Sosa et al., 2003 [84]	12	Europe	Spain	Pool	HPLC
McGuire et al., 2017 [37]	40	North America, Africa, Europe, Latin America	Ethiopia, Africa, Ghana, Kenya, Peru, Spain, Sweden, USA	Secretor/Non-secretor	HPLC
McJarrow et al., 2019 [85]	9	Middle East	UAE	Pool	HPLC-MS
Morrow et al., 2004 [35]	93	Latin America	n.a.	Pool	HPLC
Moubareck et al., 2020 [49]	18	Asia, Middle East	UAE, Iran, Oman, Yemen, Syria, India, Switzerland, UK	Pool	HPAEC
Musumeci et al., 2006 [86]	53	Africa	Burkina Faso	Pool	HPAEC
Nakhla et al., 1999 [43]	2	North America	USA	Pool, Secretor/Non-secretor	HPAEC
Nakano et al., 2001 [87]	2434	Asia	Japan	Pool	n.a.
Nijman et al., 2018 [88]	10	North America	USA	Pool	Nano-HPLC TOF
Olivares et al., 2014 [51]	24	Europe	n.a.	Secretor/Non-secretor	CE-LIF
Paganini et al., 2019 [89]	75	Africa	Kenya	Secretor/Non-secretor	HPAEC
Saben et al., 2020 [90]	136	North America	USA	Pool	HPLC
Sakaguchi et al., 2014 [91]	1	Asia	Japan	n.a.	LC-MS
Samuel et al., 2019 [92]	290	Europe	n.a.	Pool	HPLC
Sjogren et al., 2007 [93]	11	Europe	Sweden	Pool	HPLC
Smilowitz et al., 2013 [94]	52	North America	USA	Pool	NMR
Spevacek at al., 2015 [47]	15	North America	USA	Pool	NMR
Sprenger et al., 2017 [95]	34	Asia	Singapore	Secretor/Non-secretor	HPAEC-PAD
Sumiyoshi et al., 2003 [96]	16	Asia	Japan	Pool	HPLC
Thurl et al., 1996 [97]	1	Europe	n.a.	Secretor	HPAEC
Thurl et al., 2010 [98]	109	Europe	Germany	Secretor/Non-secretor	HPAEC
Tonon et al., 2019 [27]	78	Latin America	Brazil	Secretor/Non-secretor	LC-MS
Tonon et al., 2019 [99]	10	Latin America	Brazil	Pool	HPLC
Torres Roldan et al. 2020 [100]	153	Latin America	Peru	Pool	HPLC
Williams et al., 2017 [101]	16	North America	USA	Pool	HPLC-FL
Wu Wei et al., 2020 [102]	222	Asia	China	Secretor/Non-secretor	HPAEC
Zhang et al., 2019 [103]	61	Asia	China	Pool	LC-MS/MS-MRM

n.a.: not available.

**Table 4 nutrients-13-02737-t004:** Overview of the reported data for individual HMOs per lactation period.

Lactation Period	Number of Articles	Number of HMOs Quantified	Total Sample Size
Colostrum (0–5 days)	27	34	3785
Transitional (6–14 days)	20	28	3356
Mature (15–90 days)	48	36	6094
Late (>90 days)	15	28	4406

**Table 5 nutrients-13-02737-t005:** Overview of reported data for total HMOs per lactation period.

Lactation Period	Number of Articles	Total Sample Size
Colostrum (0–5 days)	8	123
Transitional (6–14 days)	5	106
Mature (15–90 days)	10	216

**Table 6 nutrients-13-02737-t006:** Descriptive statistics (g/L) of the most abundant HMOs by lactation stage in pooled globalised milk: colostrum (0–5 days), transitional milk (6–14 days), mature milk (15–90 days) and late milk (>90 days). More data available in Appendix A, Appendix A.

Colostrum (0–5 Days)	2’-FL	LNDFH-I (DF-LNT)	LNFP-I	LNFP-II	LNT	3-FL	6’-SL	DSLNT	LNnT	DFL (LDFT)	FDS-LNH	LNFP-III	3’-SL	LST c	TF-LNH
Total sample size	1101	885	1165	1027	916	833	3319	796	877	320	205	622	3391	962	96
Minimum mean	0.69	0.53	0.16	0.02	0.20	0.19	0.00	0.00	0.07	0.04	0.08	0.02	0.00	0.00	0.09
Median	2.30	0.82	0.93	0.53	0.62	0.73	0.45	0.30	0.31	0.32	0.29	0.20	0.14	0.16	0.27
Maximum mean	4.28	2.1	2.14	1.18	1.60	1.90	0.74	1.12	1.24	0.54	0.67	0.89	0.67	0.30	0.41
Mean of means	3.18	1.03	0.83	0.78	0.73	0.72	0.40	0.38	0.37	0.29	0.28	0.26	0.19	0.17	0.25
Transitional (6–14 days)															
Total sample size	789	209	798	297	645	693	5488	75	230	62	-	113	5488	194	39
Minimum mean	0.10	0.37	0.37	0.00	0.36	0.10	0.00	0.00	0.15	0.40	-	0.07	0.00	0.00	0.11
Median	2.60	1.10	1.1	0.29	0.88	0.51	0.73	0.644	0.41	0.68	-	0.4	0.13	0.488	0.23
Maximum mean	2.88	1.81	1.932	1.452	3.9	1.67	1.297	1.3	1.033	0.7	-	0.74	0.25	0.941	0.23
Mean of means	2.07	1.06	1.11	0.33	1.07	0.59	0.71	0.67	0.47	0.56	-	0.37	0.13	0.55	0.17
Mature (15–90 days)															
Total sample size	4048	2766	4156	3220	3841	3747	5691	1261	4048	3035	668	2650	6014	1549	345
Minimum mean	0.69	0.005	0.16	0.02	0.2	0.16	0.00	0.00	0.06	0.04	0.08	0.02	0.00	0.00	0.04
Median	2.3	1.074	0.93	0.539	0.62	0.73	0.45	0.3	0.31	0.32	0.29	0.2	0.14	0.159	0.18
Maximum mean	4.28	2.53	2.14	1.814	1.60	1.9	0.74	1.122	1.24	0.54	0.67	0.89	0.7	0.3	0.39
Mean of means	2.28	1.10	0.83	0.78	0.74	0.72	0.403	0.38	0.372	0.293	0.29	0.26	0.19	0.17	0.15
Late (>90 days)															
Total sample size	1951	1170	1885	990	1616	1885	4913	1014	1738	1276	71	990	3751	1227	33
Minimum mean	0.00	0.00	0.00	0.00	0.10	0.26	0.01	0.13	0.04	0.00	0.04	0.05	0.08	0.00	0.20
Median	1.72	0.67	0.43	0.35	0.56	1.18	0.19	0.20	0.19	0.27	0.16	0.22	0.13	0.04	0.20
Maximum mean	4.27	1.39	0.97	0.61	1.37	2.57	1.00	0.31	0.61	0.58	0.16	0.77	0.30	0.25	0.20
Mean of means	1.65	0.87	0.41	0.27	0.64	0.92	0.30	0.22	0.19	0.27	0.12	0.23	0.13	0.08	0.20

**Table 7 nutrients-13-02737-t007:** Overview of the records reporting total HMO data were used to calculate the average total HMO fractions throughout lactation.

Reference	Sample Size	Region	Population	Analytical Method
Albrecht et al., 2013 [104]	n.a.	Europe	Netherlands	CE-LIF-MS
Bode, 2013 [105]	n.a.	n.a.	n.a.	n.a.
Coppa et al., 1993 [106]	46	Europe	Italy	HPLC
Coppa et al., 2011 [53]	39	Europe	Italy	HPAEC
Ferreira et al., 2020 [74]	52	Latin America	Brazil	HPLC-FL
McGuire et al., 2017 [37]	40	International cohort	-	HPLC
Montreuil et al., 1960 [107]	n.a.	Europe	France	Chromatography
Newburg et al., 1995 [108]	n.a.	n.a.	n.a.	n.a.
Nijman et a., 2018 [88]	10	North America	USA	HPAEC-PAD
Viverge et al., 1985 [109]	n.a.	Europe	France	Chromatography
Viverge et al., 1990 [110]	15	Europe	France	Chromatography
Xu et al., 2017 [111]	45	Africa, North America	Malawi, USA	UPLC

n.a.: not available.

**Table 8 nutrients-13-02737-t008:** Descriptive statistics (g/L) of the total HMO fraction throughout lactation (data from 12 records).

	Colostrum (0–5 Days)	Transitional (6–14 Days)	Mature (14–90 Days)
Minimum	9.1	6.2	8.6
Median	16.7	16	10.3
Maximum	25	20	16.8
Mean	17.7	13.3	11.3
Std. Deviation	3.3	6.5	2.2

## Data Availability

Full descriptive statistics of each quantified HMO provided in a, pdf versions of each included article in the review, a Word document with detailed explanation of the pooling calculations.

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
