# Peer review of "The Mean of Milk: A Review of Human Milk Oligosaccharide Concentrations throughout Lactation"

_nutrients, 2021, doi:10.3390/nu13082737_

Round 1
Reviewer 1 Report
Over the last decades, particularly during the last few years, many data on human milk oligosaccharide (HMO) concentrations from mothers worldwide have been published. Therefore, the review submitted by Soyyilmaz et al. is a valuable and necessary contribution to the field of HMOs. In addition, this evaluation of HMO data could guide the development of infant formula.
The manuscript is well-written, the results are presented clearly and discussed convincingly. Nevertheless, as there remain several open questions particularly those concerning data processing the manuscript should be revised thoroughly.
Please, address the following specific comments.
- Line 105: Table 1: The incidence of Lewis blood groups in humans and human milk groups, respectively, is very important regarding the aim of your review article. You even calculate with these numbers. You should cite the relevant literature, favourably several articles.
- Lines 187 – 189: You included median values reported by Kunz et al. 2017. Being not a statistician, I’m not sure whether the procedure of estimating means from medians is common practice. However, there exist numerous other papers reporting median values of HMOs. Therefore, you either should exclude the study from Kunz et al. 2017 or you include all further studies reporting median values, ranges, and sample sizes.
- Lines 237-238 & Table 6: Did you consider the number of samples or the number of milk donors? You chose a weighted average approach. Does this mean that e.g., data from Azad et al. 2018 (n=446) were weighted approximately 20-fold in comparison to data from Asakuma et al. 2007 (n=20)? Generally, HMOs concentrations vary not only due to various biological effects but also because of the analytical procedures applied. It cannot be excluded, that studies reporting HMO data from many milk donors that were quantified by insufficient methods confound the outcomes of the review. Alternatively, you could ignore the number of observations (milk donors), or you choose an intermediate approach. In any case the authors should describe the calculation procedures in more detail and give reasons for their methodology.
- Line 246 Table 2 & Line 312 Table 6: For a better understanding, I recommend ordering the HMOs according to their structures instead of the relative amounts.
Furthermore, you should explain why you selected 15 HMOs for your detailed report.
- Line 249 Table 3: Please check the milk types shown in Table 3. Chaturvedi et al. 2001, Coppa et al. 2011, Nakhla et al. 1999, Thurl et al. 2010 did not examine pooled milk samples. Possibly, there are further studies with incorrect milk types.
Furthermore, the term sample should be defined. Do you mean number of milk samples or number of milk donors (mothers)?
- Line 290: Figure 2: Do you show mean values reported in Table 6? If this is true concentrations of 2’-FL and LNDFH I in colostrum don’t fit.
- Line 319: Figure 3 should be explained in more detail. Do you show sums of mean values of the 15 major HMOS reported in Table 6?
- Lines 322-330: Table 7: Several papers were cited twice, e.g. Coppa et al. 2011, McGuire et al. 2017.
- Lines 322-330: Table 7 & Table 8: Data from Coppa et al. 2011 and McGuire et al. 2017 were extracted twofold. You used concentrations of individual HMOs in order to calculate mean of means. In addition, you summed up individual HMO concentrations and showed concentrations of the total HMO fraction in Table 8. You should apply this procedure for all studies reporting the major individual HMOs by calculating the sum of all HMOs quantified for every study included. So, the mean concentrations of total HMOs would be based on many more data.
- Lines 336-338 & Figure 4 (Line 356): Here, you combine individual HMO data from the large data set of 57 studies (Table 2) with the amount of the total HMO fraction based on 12 studies indicated in Table 7. As the two data sets differ almost completely, I don’t agree with the calculation shown.
- Figure 4 (Line 357): For better eligibility colours and orders of individual HMOs should be the same in both Figures 4a and 4b.
In addition, you should indicate the structures of all HMOs, at least as supplemental material.
Author Response
Thank you so much for the general comment on the value of this paper and your elaborate review. We have added our answers in blue in the attached document. We have gone through each comment and integrated the suggestions into the manuscript.

Reviewer 2 Report
This is a systematic review article of human milk oligosaccharides. The authors report global HMO mean concentrations regardless of mother’s secretor status in order to rank order individual HMO concentrations across human milk phenotypes. The authors reports that the top 15 HMO’s in pooled human milk samples account for 75% of the total oligosaccharides in human milk. Overall the paper is quite rambling and includes data that is unnecessary and even outside of the predefined scope of this report.
- Title—What is meant by the ‘’mean of milk”; this is distracting. It is not clear from this title what exactly this paper is all about and what it contributes to the HMO literature.
- Abstract—please define secretors and non secretors; in lines 19-21, how can you evaluate length of gestation if preterm milk is excluded? ; Lines 31-22. Here you report the top ten HMO’s but the remainder of the paper consistently refers to the top 15 HMO’s. Please be consistent,
- Line 40-41. “Prolonged benefits” is an overstatement of the available data. Please rephrase: “ and associated with reduced risks for obesity, cardiovascular disease, inflammatory bowel disease and type II diabetes”. There are no RCT’s showing these benefits.
- Lines 51-52. Why would you expect human carbohydrate epitopes in animal milk?
- Line 84. Does this mean than individual HMO’s can belong to more than one structural class?
- Lines 86-103. You do not clearly state that secretors (mothers who carry Se genes) express FUT2; non-secretors(mothers who do not carry Se genes) do not express FUT2—the presence or absences of FUT2 determines secretors vs non secretors.
- Lines 140-190 ; this systematic review required significant interpretation/manipulation of cited data when the reports only listed values for secretors or non-secretors, or milk types (see Table 1); reports of pooled data also did not consistently cover the exact same lactation period. Another source of heterogeneity was the non-standardization of collection of milk samples across the data base (lines 226-228).
- Pages 6-7. Table 2 This does not really seem essential to this paper and could be reported in an on line supplement. The top 15 HMO’s are depicted here but the abstract talks about the top ten? Also line 379 that states 15 HMO’s accounted for 75% of the HMO’s. Again, consistency.
- Table 3. Tables should stand alone. The column entitled “Sample” is confusing. The values range from 1-2434. Does this represent a pooled sample from a single mother? A pooled sample from multiple mothers reported as 1? A pooled sample of 2434 mothers milk from 2434 pooled maternal samples? A pooled sample of 2434 mother’s reported as a single value? The text states that this number represents the number of donors? Why would you report a paper that studied only one mother, a single donor? Is it possible to add columns for the lactation period reported and total number of HMO’s analyzed?
- Table 4. As the authors focus on the top 115 HMO’s, the table should show the number of papers that reported the top 15 HMO’s used for the summary of this report.
- Table 5. The number of articles reporting lactation period is 23 which is less than half of the reports. How many of the 23 articles reported on more than one period of lactation? How may reports did not report lactation period? This information could be added to Table 3.
- Table 6. Both Figure 2 and table 6 are not needed. Table 6 could be in a supplementary document.
- Figure 3—this really does not add anything as 2Fl is so dominant. This could be supplementary as well.
- Lines 321-358; Tables 8, 7, and Figure 4. The collection of this data, methods, etc is not described in the methods section and involves reports not even listed in the systematic review Table 3. This seems beyond the purview of this paper. Would recommend leaving this out.
- There should be some discussion of the weaknesses of this meta-analysis including risk of bias and degree of heterogeneity of the included reports, as well as the manipulation of the cited data to produce this report
Author Response
Thank you so much for the elaborate review. We have examined the comments and integrated the changes in the manuscript. We also added a paragraph on the limitations of the study in the Discussion chapter.

Round 2
Reviewer 1 Report
Thank you for revising your manuscript which I recommend for publication.
You mentioned, that the "sum of HMOs" and "total HMOs" are principally not identical. However, some authors do not differentiate and equal "sum of HMOS" with "total HMOs". In your report you calculated that this difference would be less that 10% with mass concentrations and less than 5 % with molar concentrations.
In addition, I would have discussed more extensively why you decided to take into account the number of milk donors at the expense of number of studies when calculating means.
Reviewer 2 Report
No further comments.